# Effects of Ag-Rich Nano-Precipitates on the Antibacterial Properties of 2205 Duplex Stainless Steel

Je-Kang Du [1,2], Chih-Yeh Chao [3,†], Lin-Lung Wei [4], Chau-Hsiang Wang [2], Jeng-Huey Chen [1,2,‡], Ker-Kong Chen [1,2,*] and Ruei-Bin Huang [3]

1    Department of Dentistry, Kaohsiung Medical University Hospital, Kaohsiung 807, Taiwan; dujekang@gmail.com (J.-K.D.); 720101@kmu.edu.tw (J.-H.C.)
2    School of Dentistry, College of Dental Medicine, Kaohsiung Medical University, Kaohsiung 807, Taiwan; 730119@kmuh.org.tw
3    Department of Mechanical Engineering, National Pintung University of Science and Technology, Pingtung 91201, Taiwan; cychai@mail.npust.edu.tw (C.-Y.C.); knocker546b@gmail.com (R.-B.H.)
4    Department of Materials Science and Engineering, National Chiao Tung University, Hsinchu 300, Taiwan; LLWei@jiedong.com.tw
*    Correspondence: enamel@kmu.edu.tw
†    Equal first author.
‡    Equal correspondence.

**Abstract:** The effects of the addition of silver on the microstructural variation and antibacterial performance of 2205 duplex stainless steel after solution and aging treatment were investigated by scanning electron microscopy (SEM), transmission electron microscopy (TEM), high-resolution TEM, and antibacterial testing. The microstructure showed that 2205Ag is composed of a ferrite ($\alpha$) + austenite ($\gamma$) duplex phase and Ag-rich nano-precipitates (Ag-NPs). The morphology of the Ag-NPs varied from spherical to polygonal after aging treatment at 450 °C for 4 h. These precipitates were identified as face-centered-cubic structures with a lattice parameter of a = 0.354 nm and a mismatch of $\delta$ = 0.84% relative to the austenite matrix. Notably, 2205Ag with polygonal Ag-NPs exhibited excellent antibacterial properties that were superior to those of 2205Ag with spherical Ag-NPs.

**Keywords:** duplex stainless steel; nano-precipitates; antibacterial performance

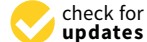



## 1. Introduction

Stainless steel is widely used in medical applications due to its high corrosion resistance [1–5]. Recently, there has been a trend of using duplex stainless steel (DSS) as a replacement for austenitic stainless steel because the former has significantly higher stress corrosion cracking (SCC) resistance than the latter [6]. Among DSSs, 2205 DSS is a promising candidate for use in medical devices due to its good corrosion resistance, mechanical properties, and significantly higher passivation range compared to those of the commonly used 316L SS [7]; 2205 DSS is thus better suited for the fabrication of orthodontic brackets and orthodontic mini-screws [8], particularly when it is used in conjunction with either stainless steel or titanium alloy arch wires [9].

The last two decades have seen a rapid increase in the development and widespread use of DSSs, with numerous studies investigating their microstructure [10–18] and generally identifying them as having a ratio of 44:56 of the austenite ($\gamma$) and ferrite ($\alpha$) phases [19]. Additionally, the $\sigma$ phase can be formed along with precipitates such as $M_{23}C_6$, $\chi$, $\sigma$, $\gamma'$, or $Cr_2N$, either within the matrix and/or on grain boundaries after an aging treatment [20,21]. Furthermore, these precipitates can also influence the corrosion resistance and mechanical properties of DSSs depending on the alloy composition and its thermal treatment history [22–26].

Although stainless steel is widely used in various medical applications, it lacks antibacterial properties. Thus, bacterial infection occurs on biomedical implants, which affects

the postoperative infection and failure rates and incurs high medical expenses [27,28]. The growth of bacteria on medical equipment is the first step leading to infection and inflammation. The bacteria gathered while in contact with the surrounding tissues are prone to forming biofilms that can cause infection and increase clinical complications. For instance, it was found that several cases of failure of orthodontic mini-screws used in dental clinical practice were due to inflammation and infection in the oral cavity [29,30]. Furthermore, bacterial adhesion causes metal pitting and crevice corrosion, resulting in degraded mechanical properties of orthodontic mini-screws and leading to treatment failure [31].

As microbial adhesion is a common source of contamination for stainless-steel-based medical devices, recent studies have focused on the development of new antibacterial materials [32]. In one approach, surface modifications were used to reduce and avoid the attachment and growth of bacteria on medical devices. These modifications, particularly the incorporation of Ag or Cu elements, are a viable method for improving the suitability of SS for biomedical applications and can be carried out via electroplating/electrodeposition [33,34], spraying [35], ion implantation, and plasma [36–42]. However, the antibacterial effect may be lost due to surface cleaning, friction, process, time, and usage [43]. In an alternative approach, surface modification methods such as coating and spraying with alloys with antibacterial ability have been developed recently. For example, Xi et al. [44] showed that the antibacterial property of Cu-bearing 316L SS increased with the formation of evenly distributed Cu-rich precipitates in the matrix after aging treatment at 700 °C for 6 h. Other studies [45–47] also indicated that the formation of nano-sized Cu-rich or Ag-rich precipitates in SS or Ti alloys can play a key role in the development of antibacterial properties. The presence of the precipitates in the metal implies that alloys containing copper or silver after heat treatment can retain their antibacterial ability after grinding.

In the food industry, silver is added to stainless steel because it is non-toxic and has an antibacterial effect [48]. As reported in previous studies on the antibacterial effects of silver, silver-containing antibacterial stainless steel exhibits a strong antibacterial effect and does not have a harmful effect on the human body [49]. The antibacterial mechanism of silver is due to the interaction of silver ions with cells. The negatively charged cell wall is the strongest outer layer in bacteria, and silver ions are positively charged. When the silver ions interact with the wall to produce electrical neutralization, the cell wall charge is weakened, leading to a hole in the cell wall through which the intracellular tissue fluid flows out, preventing the growth and reproduction of the bacterium. The release of Ag ions can increase cellular oxidative stress [50] and disturb membrane protein activities. In addition, the negative charge of the bacterial membrane attracts the positive charge of Ag ions to adhere to the bacterial wall, causing penetration of the bacterial membrane and the bacterium's death [51]. Recently, some studies have indicated the mechanisms of interaction between $Ag^+$ ions and bacterial membranes: (1) interaction between $Ag^+$ ions and external peptidoglycans can resist oxygen passing into the bacterial cell [52]; and (2) interaction between $Ag^+$ ions and thiol group proteins can inactivate bacterial enzymatic functions [53] to damage DNA [54] and the cell envelope and cytoplasm due to vacuole deposition [55,56]. In order to achieve the antibacterial mechanisms of $Ag^+$ ions, the Ag concentration added to materials must be sufficiently high [57]. Previous studies revealed that materials containing at least 3 wt% Ag could effectively obtain antibacterial properties [58–60]. However, Ag-containing materials may have a detrimental influence on human health, such as by affecting testicular function [61], deceasing weight [62], and causing blood biochemical inflammation [62]. Therefore, it is important to decrease the Ag concentration and confirm antibacterial effects at low Ag concentrations. Pavel et al. [53] indicated that low $Ag^+$ concentration can lead to massive proton leakage to promote bacterial death.

Currently, no stainless-steel alloys with an antibacterial effect are commercially available. Accordingly, the aim of this study was to fabricate bulk 2205 DSS containing a low Ag concentration with the addition of 0.15 wt% silver. Furthermore, the effect of heat treatment

for silver-containing 2205 DDS on its antibacterial properties against *Escherichia coli* (*E. coli*) and *Staphylococcus aureus* (*S. aureus*) was investigated.

## 2. Materials and Methods

### 2.1. Sample Preparation

The 2205 and 2205Ag DDS were prepared in an air induction furnace from powders of iron (99.5% purity), chromium (99.9%), molybdenum (99.9%), pure silver, and 6% nitrogen–chromium iron. The chemical composition of the alloy used in this study is listed in Table 1. After homogenization at 1150 °C for 2 h in a protective argon atmosphere to eliminate segregation, the ingot was hot-forged to a thickness of 3.0 mm and cut into $40 \times 40$ mm$^2$ pieces for further experiments. 2205 (named 2205-S1) and 2205Ag were subsequently solution-treated at 1080 °C for 30 min and then quenched in water. Finally, the specimens were aged at 450 °C for 1 or 4 h to precipitate the Ag-rich phase (named 2205Ag-A1 and 2205Ag-A4).

**Table 1.** Chemical composition of the 2205 and 2205Ag used in this study (mass%).

| Elements | Fe | Cr | Ni | Mo | Mn | C | N | Si | Ag |
|---|---|---|---|---|---|---|---|---|---|
| 2205 | Bal. | 23.44 | 5.90 | 3.07 | 1.12 | 0.03 | 0.10 | 0.31 | <0.01 |
| 2205Ag | Bal. | 22.51 | 5.45 | 2.82 | 1.25 | 0.03 | 0.18 | 0.44 | 0.15 |

### 2.2. Microstructure Characterization

The specimens were ground with SiC abrasive paper to grade #2000 and polished with 0.3-μm $Al_2O_3$ powder, then etched in a solution containing 2.5% HCl, 7.5% $HNO_3$, and 90% glycerol. Microstructural analysis was performed by scanning electron microscopy (SEM, JSM-6380, JEOL, Tokyo, Japan) at 15 kV.

The specimens for transmission electron microscopy (TEM) were prepared using a double-jet electropolisher operating at $20 \pm 2$ V with an electrolyte consisting of 70% ethanol, 20% acetic acid, and 10% perchloric acid; they were then stored at a temperature between $-20$ and 0 °C. Observation of the samples was performed via TEM (CM200, PHILPLIES, Tokyo, Japan) at 200 kV and high-resolution TEM (HR-TEM, ARM200F, JEOL, Tokyo, Japan) providing Cs = 0.5 mm and an atomic resolution of 0.08 nm at 200 kV. Crystallographic analysis of the matrix and precipitates was investigated by way of selected area diffraction patterns (SADPs) and energy-dispersive X-ray spectrometry (EDS) measurements to determine the composition of each phase.

### 2.3. Antibacterial Testing

Antibacterial testing was performed in accordance with the Japanese Industrial Standard (JIS) Z 2801:2000 (Japanese Standards Association, 2000) [63], which is a widely used method for assessing the antimicrobial ability of different materials. For this test, all of the specimens and tools were sterilized at 121 °C for 15 min in an autoclave. Then, *E. coli* (ATCC-25922) and *S. aureus* (ATCC-25923) at a density of $10^6$ CFU/mL in suspension (0.4 mL) were individually dripped onto the control group and each specimen (surface area of $10 \times 10$ mm$^2$) and incubated at 37 °C for 24 h. All of the bacteria on the specimen surface were rinsed with phosphate-buffered saline (PBS) repeated three times to ensure bacteria were removed from the surface. The suspension was diluted (up to $10^{-6}$) in PBS. Then, the washing solution (0.1 mL) was spread onto lysogeny broth (LB) agar plates and incubated at 37 °C for 24 h to calculate the number of colonies (CFUs/mL). All experimental procedures were repeated in triplicate, and the obtained average values were calculated. The following formula was used to calculate the antibacterial rate (AR) of each specimen:

$$AR = (N_{2205} - N_{specimens})/N_{2205} \times 100\%, \tag{1}$$

where $N_{2205}$ and $N_{specimens}$ are the average numbers of bacterial colonies on the 2205 and 2205Ag specimens, respectively.

## 2.4. Ag Ion Release

The amount of released $Ag^+$ ions was measured by inductively coupled plasma mass spectrometry (ICP–MS, PerkinElmer Inc.—Optima 2100 DV, Waltham, MA, USA). The immersion was conducted according to ISO Standard 10993-15:2000. The samples were individually immersed in a biological environment (Dulbecco's modified Eagle medium (DMEM, St. Louis, MO, USA) with serum) at 37 °C. One milliliter of the collected solution was used to analyze the Ag ion concentration after 1, 3, and 5 days. The experiments were repeated three times.

## 2.5. In Vitro Cytocompatibility

The in vitro cytotoxicity of the 2205 samples was evaluated using mouse embryonic fibroblast cells (NIH/3T3 cells, Food Industry Research and Development Institute, Hsinchu, Taiwan)) via MTT (3-(4,5-dimethyl-2-thiazolyl)-2,5-diphenyltetrazolim bromide, MTT, Sigma, St. Louis, MO, USA) assay according to ISO 10993-5. NIH/3T3 cells were cultured in medium containing DMEM (Sigma, St. Louis, MO, USA), 10% fetal bovine serum, and 1% penicillin–streptomycin in an environment of 37 °C and 5% $CO_2$ atmosphere. All samples were autoclaved at 121 °C for 40 min. Each specimen ($n$ = 3) was extracted in medium for 1, 3, and 5 days at 37 °C. Cells at a density of $1.0 \times 10^4$ cells/well were seeded in a 96-well plate and cultured for 24 h. After incubating for 24 h, the medium was removed, and then 100 μL of 100% extraction from each specimen was added into each well. After culturing with the extraction for 24 h, the extraction was replaced by adding 100 μL of the MTT solution, and cells were incubated for another 2 h at 37 °C. Subsequently, the medium was removed and 200 μL dimethyl sulfoxide (DMSO) was added into each well to dissolve the MTT formazan. After 20 min, the optical density (OD) was measured by an ELISA reader (Infinite F50, Tecan, Männedorf, Zürich, Switzerland) with a wavelength at 570 nm. The whole experiment was repeated three times. The relative growth rate (RGR) of the NIH/3T3 cells was considered as indicating non-cytotoxicity when the value was higher than 75%. The RGR was calculated by using the following formula:

$$RGR = (OD_{control} - OD_{specimens})/OD_{control} \times 100\%. \tag{2}$$

## 3. Results and Discussion

### 3.1. Microstructures

The microstructure of the 2205-S1 was essentially that of the ferrite ($\alpha$) and austenite ($\gamma$) duplex phases after the solution treatment at 1080 °C for 30 min, as shown in Figure 1a. Figure 1b shows that the Ag-modified 2205Ag-A1 retained its ferrite–austenite ($\alpha + \gamma$) duplex microstructure. After aging treatment at 450 °C for 4 h, numerous silver nano-precipitates (Ag-NPs) were formed within the austenite matrix. The diameter of these granular NPs was approximately 30–60 nm. The low melting point of silver (961 °C) aided the formation of liquid silver in the microstructure during the homogenization process at 1150 °C. Then, solidified liquid silver caused the dispersion of Ag-NPs in the 2205Ag DSS, as shown in Figure 1d. In this study, it was found that the Ag-NPs were located at the ferrite/austenite interface. In addition, Yang et al. [46] found that Ag-NPs were not only formed at the $\alpha/\gamma$ boundaries but also at the ferrite phase and the austenite phase. Nevertheless, their analysis showed that the probability of Ag particle formation was higher at the $\alpha/\gamma$ boundaries than at other locations.

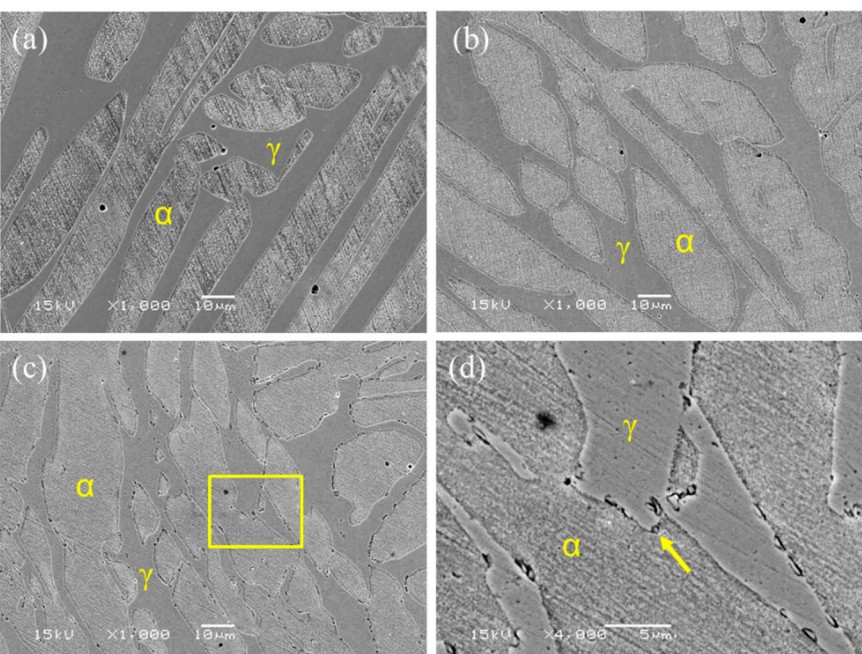

**Figure 1.** SEM images of the 2205 duplex stainless steel (DSS) specimens. (**a**) 2205 after solution treatment, indicating the α and γ duplex phases; (**b**) 2205Ag after 450 °C aging treatment for 1 h, indicating the α and γ duplex phases; (**c**) 2205Ag after 450 °C aging treatment for 4 h; (**d**) a magnified view of the region marked by the yellow box in (**c**), indicating the α, γ, and silver nano-precipitate (Ag-NP) phases (yellow arrow).

Furthermore, no precipitates were observed in the bright field (BF) TEM micrograph of the 2205-S1 specimen, as shown in Figure 2a. Figure 2b,c shows two SADPs obtained from the regions marked A and B in Figure 2a, respectively. These regions were identified as the FCC (γ) phase and BCC (α) phase for the [011] and [011] planes of the foil, respectively. The lattice parameters of the γ and α phases were a = 0.360 nm and a = 0.276 nm.

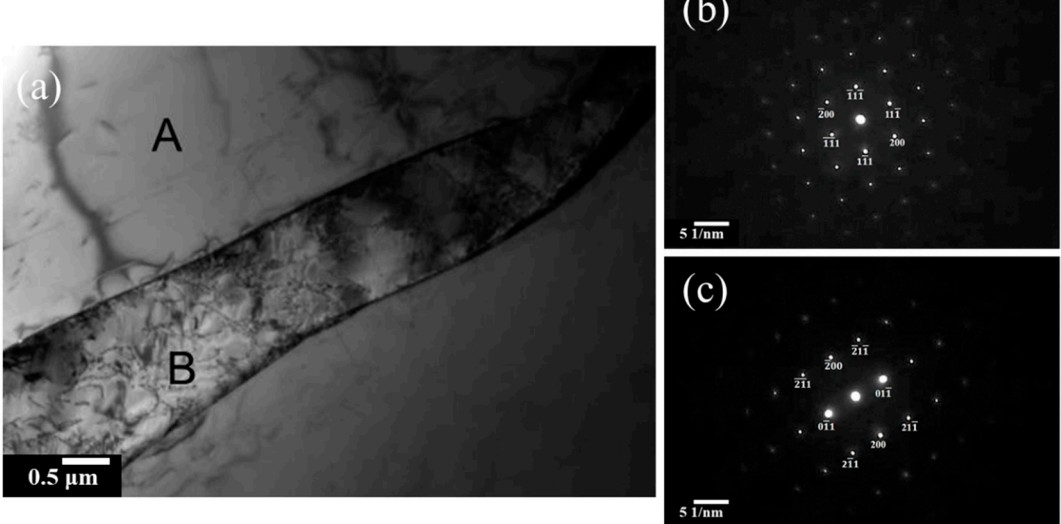

**Figure 2.** TEM micrographs of 2205 after solution treatment: (**a**) bright field (BF) image revealing the γ (region A) and α (region B) duplex phases; (**b**) selected area diffraction pattern (SADP) of the matrix marked as A in (**a**) where the foil normal is [011]; and (**c**) SADP of the matrix marked as B in (**a**) where the foil normal is [011].

However, after aging at 450 °C for 1 h, some precipitates were formed within the austenite matrix, as shown in Figure 3a. The size of these particles was approximately 15 nm, with the typical morphology shown in the HRTEM image in Figure 3b. EDS data obtained from the fine precipitate indicated that it comprised Ag-NPs, as shown in Figure 3c.

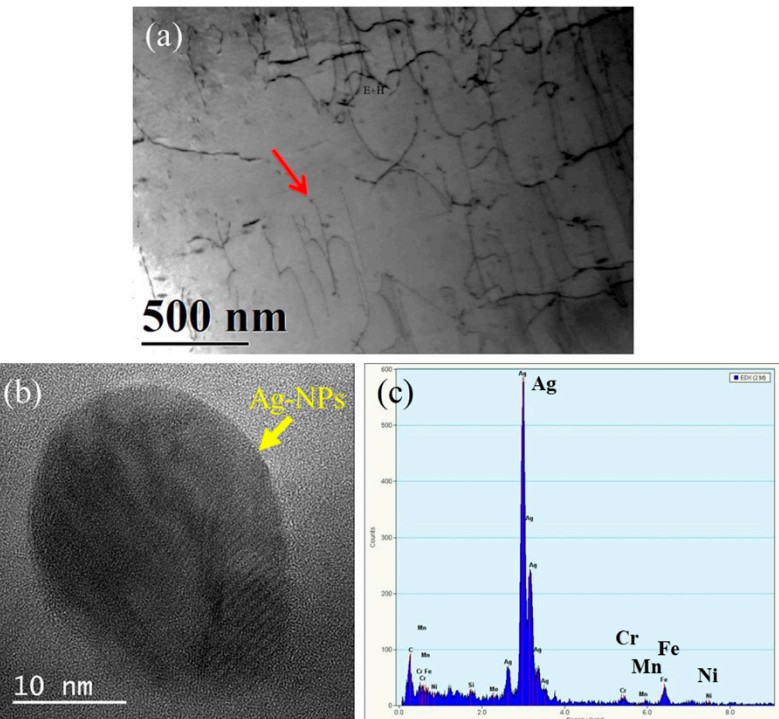

**Figure 3.** TEM micrographs taken from the 2205Ag after 450 °C aging treatment for 1 h. (**a**) BF image showing some fine precipitates (red arrow) within the austenite phase, (**b**) HR-TEM image of an Ag-NP, showing that the diameter of Ag-NPs was about 15 nm; and (**c**) EDS analysis results for the Ag-NP.

An increase of the aging time to 4 h resulted in an increase in the number of Ag-NPs, as shown by the bright and dark field images (red arrows) presented in Figure 4a,b, respectively. To further analyze the nanoparticles, the samples were examined by HR-TEM imaging in order to provide a detailed characterization of the interface between the nanoparticle and austenite matrix. Figure 4c shows the HR-TEM images obtained from the same specimen, indicating that the nanoparticle morphology tended toward polygonal with some interfacial dislocations, and the particle size was increased to approximately 30 nm. Figure 4e shows a magnified view of the region marked by A in Figure 4c, revealing that a (110) atomic arrangement and a coherent interface were created between the matrix and Ag-NPs. An examination of the diffraction pattern of the same region shown in Figure 4d revealed that the precipitates had a face-centered cubic (FCC) structure with lattice parameter a = 0.354 nm, while the small mismatch ($\delta$ = 0.84%) indicated that their interface with the matrix was fully coherent at the nanoscale. In addition, the TEM-EDS line-scan results show that the Ag content was higher in the Ag-NPs (approximately 40.8 at %) than in the matrix (Figure 3a).

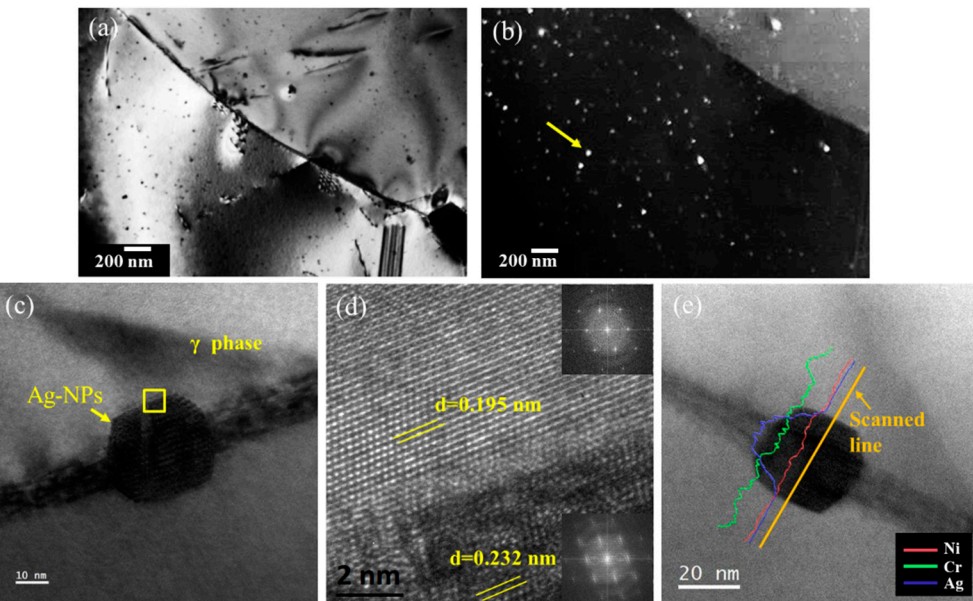

**Figure 4.** TEM and HR-TEM micrographs of the 2205Ag after 450 °C aging treatment for 4 h. (**a**) BF image; (**b**) DF image showing some fine precipitates (indicated by yellow arrows) within austenite; (**c**) HR-TEM image of a polygonal Ag-NP; (**d**) HR-TEM image of the region marked by the yellow box in (**c**); and (**e**) results of line-scan EDS analysis for the polygonal Ag-NP, showing that the Ag-NP had high Ag content.

Due to the low solubility of Ag dissolved in Fe in liquid states, the solubility of Ag in solid Fe presented a maximum of 0.022 at% at 1398 °C in the austenite phase [64]. Yang et al. [65] showed that after homogenization at 1120 °C, the solubility of silver increased in both the austenite and ferrite phases and reached $0.10 \pm 0.02$ and $0.09 \pm 0.01$ wt%, respectively. However, as the aging treatment temperature decreased, the solubility of Ag in the austenite and ferrite phases decreased slightly. This demonstrated that the Ag precipitates formed more easily during aging at low temperatures. In the present study, an aging treatment at 450 °C was used to obtain Ag-rich precipitates, as shown in Figure 4b. It was observed that the morphology of the Ag-NPs changed from spherical to polygonal after the heat treatment. This may be because aging treatment provides limited energy for diffusion and growth of Ag-NPs in the matrix. The growth of Ag-NPs within a diffusion-limited regime (relatively low energy) resulted in the faceted morphology of the resulting nanocrystals [66].

Yang [46] showed that 2205-0.2Ag DSS had a typical microstructure of $\alpha$-ferrite, primary $\gamma$-austenite ($\gamma$), secondary $\gamma$-austenite ($\gamma'$), and circular Ag-rich particles after high-temperature aging (1000 °C and 1100 °C). Additionally, $\alpha$-ferrite transferred to the $\gamma'$, $\chi$, $\sigma$, and $Cr_2N$ phases when 2205-0.2Ag DSS underwent 800 °C and 900 °C aging treatment. Further, the solubility of Ag in the $\alpha$, $\gamma$, and $\gamma'$ phases decreased progressively with decreasing aging temperature. This implied that lower aging temperature would progress the Ag particle precipitation. Owning to the solubility of Ag in Fe being very low, the Ag-rich particles could be found after various aging treatments at the $\alpha$ phase, $\gamma$ phase, and $\alpha/\gamma$ boundaries.

### 3.2. Effect of Ag-NPs on Antibacterial Properties

The numbers of *E. coli* and *S. aureus* colonies present on the surface of the agar plate following bacterial adhesion testing of the samples obtained using different alloy treatments are shown in Figure 5. The results revealed that there was little difference in the behavior of the two types of bacteria and that the addition of 0.15 wt% Ag produced only a slight antimicrobial effect. However, after the alloy was aged at 450 °C, its antibacterial performance increased significantly, as can be inferred from the sharp decrease in the number of bacterial colonies. The antibacterial rates of 2205Ag-A4 against *E. coli* and

*S. aureus* were 86.1% and 82.9%, respectively. This suggests that a further increase in the number of Ag-NPs may allow for an additional improvement in its antimicrobial efficacy.

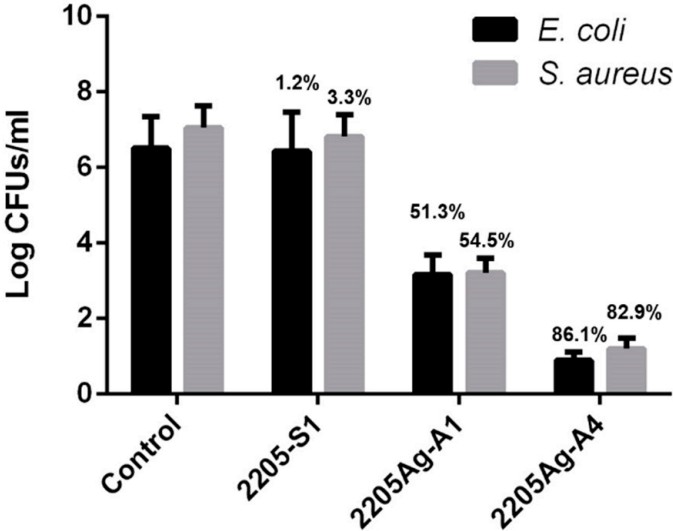

**Figure 5.** Standard plate count of bacteria (CFU) and the corresponding antibacterial properties (%) of the 2205-S1, 2205Ag-A1, and 2205Ag-A4 specimens against *Escherichia coli* and *Staphylococcus aureus*.

In this study, each specimen had a quantitative bacterial culture, and it was found that 2205Ag-A4 had strong antibacterial properties that can be attributed to the polygonal Ag-NPs with greater release characteristics of Ag ions. Nel et al. [67] proposed that the physical and chemical functions of NPs mainly depend on the size and shape of the particles (which can be spherical, triangular, tubular, polygonal, or even irregular), the presence or absence of crystal structure, surface energy, and solubility. Some researchers have also suggested that the size and shape of Ag-NPs affect their antibacterial ability. Compared to ordinary Ag-NPs, triangular and polygonal Ag-NPs will provide greater antibacterial ability, mainly because the triangular shape makes metals more prone to dissociation due to the higher surface energy of the undercoordinated atoms at the sharp edge, leading to a faster release of Ag cations [68].

Previous studies have also shown that silver ions have excellent antibacterial effects against *E. coli* and *S. aureus* [38,46,69]. It has been proven that the antibacterial property of stainless steel is due to the dissolution of positive ions due to the galvanic effect of the Ag in the metal matrix [69]. The antibacterial ability is also positively related to the amount of released ions. This is because Ag ions can react with either amino acids or amino groups ($-NH_2$) of bacterial membranes and the enzymes produced by bacterial metabolism inside the bacterial cells, eventually killing the bacteria. The presence of bacteria can also lead to increased dissolution of Ag from the surface of the Ag-bearing material through Ag complex formation, leading to effective and wide-spectrum bacterial inhibition [70]. Thus, our results are in agreement with the studies previously reported in the literature.

When the size of the silver particles is less than 100 nm, these particles are considered to be Ag-NPs. Anna et al. prepared Ag-NPs with a size of 5–12 nm for antibacterial testing and showed that Ag-NPs with a diameter of 5 nm had the best antibacterial performance. It was also confirmed that the antibacterial performance was enhanced with decreasing size of the Ag-NPs [71]. Ag-NPs with size in the 5–60 nm range show significant antibacterial ability, and, therefore, Ag-NPs in this size range have mainly been investigated in recent and current studies [72]. Another alternative for the modification of stainless steel by Ag-NPs is the chemical assembly of Ag-NPs on the steel surface; this can be achieved by using a coupling agent such as 3-aminopropyltriethoxysilane (APTES) [73].

### 3.3. Cytocompatibility Evaluation

It is known that Ag-NPs oxidize and form Ag⁺ [74]. Thus, we should observe the effects of Ag ions from Ag-NPs. Figure 6 shows the cumulative Ag ion concentration released from the 2205Ag samples after immersion in DMEM for 1, 3, and 5 days. The data revealed that the concentration of released Ag ions increased with immersion time [75]. Further, 2205Ag-4A exhibited a relatively higher concentration than 2205Ag-1A, indicating that the enhanced antibacterial property of 2205Ag-4A was contributed by the released Ag ions. The release concentration of silver ions in a physiological environment showed that $58.8 \pm 7.6$ µg/L accumulated over five days (11.8 µg/L/day). Additionally, the cytotoxicity test (Figure 7) showed that the 2205Ag-1A and 2205Ag-4A exhibited over 75% cell viability, indicating non-toxicity with increasing amounts of released Ag ions. In addition, there was no significant difference between the cell morphologies after culturing with the extraction in each group and the control group (as shown in Figure 8). This indicates that even though the release of silver ions increased after the heat treatment, it still did not present toxicity against the cells.

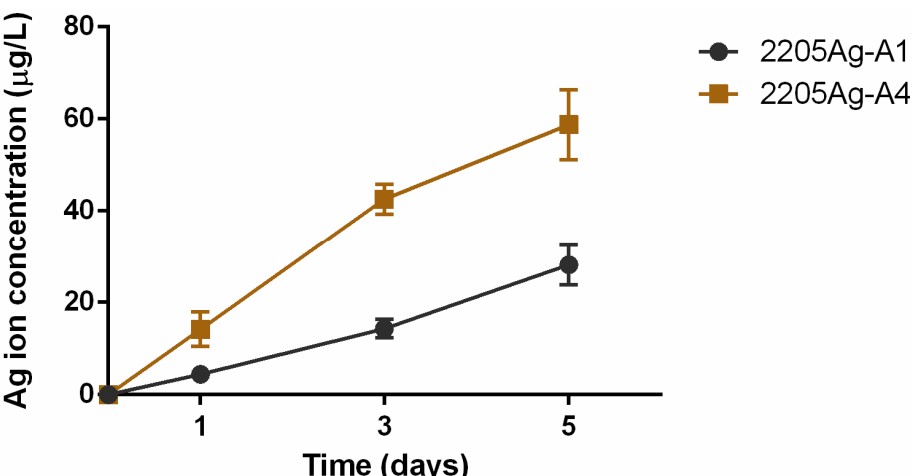

**Figure 6.** Effects of time on Ag ion release from 2205Ag specimens after immersion in Dulbecco's modified Eagle medium (DMEM) for 1, 3, and 5 days.

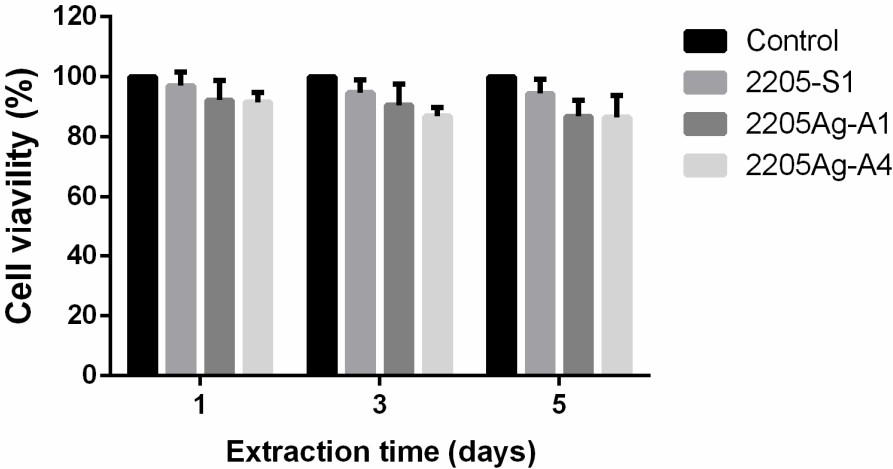

**Figure 7.** Cytotoxicity test after culturing with different extraction times from different specimens for 24 h.

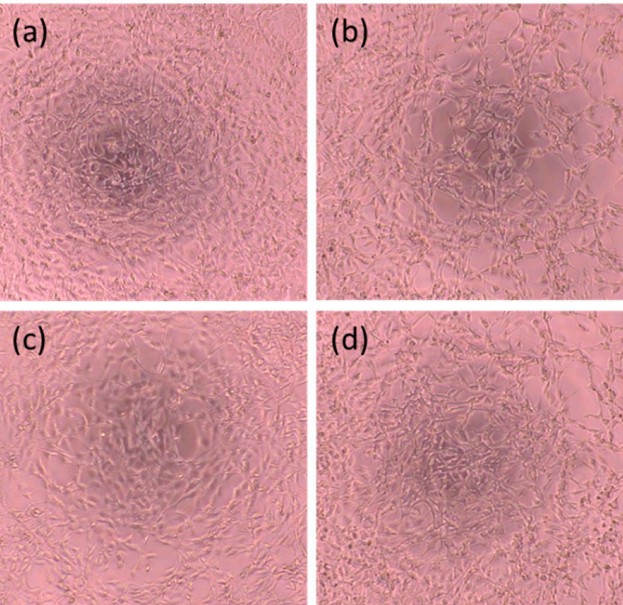

**Figure 8.** Morphology of NIH/3T3 cells after culturing with different periods of extraction from 2205Ag-4A for 24 h. (**a**) Control; (**b**) 1 day of extraction; (**c**) 3 days of extraction; and (**d**) 5 days of extraction.

Additionally, some antibacterial properties can be imparted to DSS via the deposition of nanoparticles into the pores on its surface [76]. For example, ion implantation of Ag or Cu onto stainless steel is known to significantly improve its antibacterial activity because the dissolution of these ions from the surface generates a bactericidal effect [36–39]. The antibacterial activity provided by surface treatment is significantly long-lasting, and the surface often maintains an antibacterial rate of >90% for over a year even after exposure to air or water [38]. However, the steels modified by such methods may release too high a content of Ag ions after a long immersion time [38,77]. This implies that although it is possible to coat the surface of DSS with Ag to achieve antibacterial properties, the strength of the chemical anchoring of the nanoparticles to metals is a challenging issue because of its impact on the ion release rate that, in turn, is related to toxicity. In addition, the antibacterial action will invariably decrease during the life of any medical device. Thus, it is necessary to achieve an appropriate balance between continuous and effective antibacterial functions and biological safety. Therefore, it is important to determine whether steel can remain non-toxic while maintaining antibacterial activity due to the application of NPs.

Jamuna-Thevi [78] studied the cytotoxicity and antibacterial activity of 1~4 wt% Ag-containing $TiO_2$ thin-film coatings on SS. They showed that silver ion release was greater in the early stage and decreased with time. They demonstrated that 2 wt% Ag-containing $TiO_2$ thin-film coatings could release about 30~95 ppb Ag ions in the first day, which could provide an antibacterial reduction rate of over 99% with no toxicity to cells. This implies that the release of ions needs to be at least about 30 ppb to achieve excellent and safe antibacterial performance.

The Ag ions released from the surface of 2205Ag were measured at between 5.6 and 11.8 ppb/day, which achieved the minimum concentration for antibacterial function (0.1 ppb) [78]. Additionally, this value was also less than the maximum concentration of 10 ppm that human cells can tolerate without a toxic reaction. The daily amount of normal Ag ions that the human body can consume is about 200 ppb [78]. However, when the Ag ion concentration exceeds 300 ppb in the blood, side effects in the human body such as liver and kidney damage can occur [79].

## 4. Conclusions

In this study, we investigated the microstructures and antibacterial properties of 2205 DSS modified by the addition of 0.15 wt% Ag. The microstructure of the Ag-bearing 2205 DSS was a mixture of the $\alpha$, $\gamma$, and Ag-NP phases. After aging at 450 °C for 4 h, the morphology of the Ag-NPs varied from spherical to polygonal. The polygonal Ag-NPs comprised an FCC structure with a lattice parameter of a = 0.354 nm; however, their small lattice mismatch ($\delta$ = 0.84%) with the austenite matrix ensured a fully coherent interface. It was found that the alloy with polygonal Ag-NPs showed better antibacterial properties against *E. coli* and *S. aureus.*

**Author Contributions:** Conceptualization, C.-H.W., K.-K.C. and C.-Y.C.; methodology, J.-K.D. and C.-Y.C.; software, L.-L.W. and R.-B.H.; validation, J.-K.D. and J.-H.C.; formal analysis, C.-Y.C. and K.-K.C.; investigation, J.-K.D. and L.-L.W.; resources, J.-H.C. and C.-H.W.; data curation, J.-H.C. and R.-B.H.; writing—original draft preparation, J.-K.D. and C.-Y.C.; writing—review and editing, J.-K.D., R.-B.H. and K.-K.C.; visualization, K.-K.C.; supervision, C.-H.W.; project administration, J.-H.C.; funding acquisition, J.-K.D. and C.-H.W. All authors have read and agreed to the published version of the manuscript.

**Funding:** This study was financially supported by funding from the Ministry of Science and Technology, MOST106-2314-B-037-013- and MOST99-2314-B-037-073-MY3, and Kaohsiung Medical University Hospital, KMUH105-5R63 and KMUH106-6R73.

**Institutional Review Board Statement:** "Not applicable" for studies not involving humans or animals.

**Informed Consent Statement:** "Not applicable" for studies not involving humans.

**Data Availability Statement:** Data sharing not applicable.

**Conflicts of Interest:** The authors declare no conflict of interest.

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
