# Peer review of "Effects of Ag-Rich Nano-Precipitates on the Antibacterial Properties of 2205 Duplex Stainless Steel"

_metals, doi:10.3390/met11010023_

Round 1
Reviewer 1 Report
The paper is very interesting and the research topic is of great interest.
I suggest to add some tests as here described.
Introduction:
1) "The negatively charged cell wall is the strongest outer layer in bacteria, and
silver ions are positively charged": I suggest to consider and discuss the difference between the mechanism of action of silver ions in a ionic compound and silver atoms in an alloy.
2) Results: "with the higher release characteristics of Ag ions ... leading to a faster release of Ag cations ". Actually, there is no data concerning ion release from the tested Ag doped SS. I suggest to perform these tests both in inorganic and biological (serum with proteins) fluids. It is of interest to compare the obtained values of ion release with the published thresholds for bacterial effects (which must be overcome) and cytotoxicity (which must not be overcome - see below).
3) "Thus, it is necessary to achieve an appropriate balance between continuous and effective antibacterial functions and biological safety". Actually, I agree that this point is critical and crucial. The scientific literature is rich of antibacterial surfaces, but a lot of them are cytotoxic. I suggest to add a biocompatibility test with fibroblasts or osteoblasts, too.
Author Response
19/12/2020
Prof. Dr. Hugo F. Lopez
Editor-in-Chief
Metals
Dear Prof. Dr. Hugo F. Lopez:
We would like to resubmit our revised manuscript, titled " Effect of Ag-rich Nano-Precipitates on the Antibacterial Properties of 2205 Duplex Stainless Steel. " (Manuscript ID: metals-1023440), for consideration for publication in Metals.
The editor and reviewers have recommended major revisions. We have carefully reviewed these suggestions and revised our text accordingly. Please note that those words within manuscript, tables and figures are written with red words for correction. Our responses to each of the reviewers’ comments are detailed below. The details are provided in the manuscript and mark with red for correction.
Comments and Suggestions for Authors:
Reviewer 1 :
Comments and Suggestions for Authors
The paper is very interesting and the research topic is of great interest.
I suggest to add some tests as here described.
Introduction:
- "The negatively charged cell wall is the strongest outer layer in bacteria, and silver ions are positively charged": I suggest to consider and discuss the difference between the mechanism of action of silver ions in a ionic compound and silver atoms in an alloy.
Answer: We appreciate this helpful comment from the reviewer. The manuscript also revised. This information has been supplemented at line 76-91 Page 2 of revised manuscript and marked with red.
- Results: "with the higher release characteristics of Ag ions ... leading to a faster release of Ag cations ". Actually, there is no data concerning ion release from the tested Ag doped SS. I suggest to perform these tests both in inorganic and biological (serum with proteins) fluids. It is of interest to compare the obtained values of ion release with the published thresholds for bacterial effects (which must be overcome) and cytotoxicity (which must not be overcome - see below).
Answer: Thank you for your suggestion. To understand the amount of released Ag, the ICP-MS was used to analyze. The specimens were immersed in the DMEM (biological fluids) for certain days according to ISO 10993. Also, the data were compared the obtained values. In which, it showed that the released Ag ion of obtained 2205 Ag did not exceed the maximum of the daily consume in human body. This information has been supplemented at line 281-309 and Fig 6~8. Page 8~10 of revised manuscript and marked with red.
- "Thus, it is necessary to achieve an appropriate balance between continuous and effective antibacterial functions and biological safety". Actually, I agree that this point is critical and crucial. The scientific literature is rich of antibacterial surfaces, but a lot of them are cytotoxic. I suggest to add a biocompatibility test with fibroblasts or osteoblasts, too.
Answer: Thank you for your deep concern. To understand the biological safety, the biocompatibility of the 2205Ag was test by NIH/3T3 cells with extraction which was prepared by each specimen at different time points. After testing, it is confirmed that the alloy in this study can maintain its biocompatibility even with a certain antibacterial ability. This information has been supplemented at line 281-309 and Fig 6~8. Page 8~10 of revised manuscript and marked with red.
Thank you for your consideration. We hope our manuscript is suitable for publication in your journal.
Sincerely,
Professor
Ker-Kong Chen
Department of Dentistry,
Kaohsiung Medical University Hospital,
Kaohsiung 807, Taiwan, ROC.
enamel@kmu.edu.tw

Reviewer 2 Report
The manuscript entitled “Effect of Ag-rich Nano-Precipitates on Antibacterial Properties of 2205 Duplex Stainless Steel” (metals-1023440)” is an interesting study about the effects of the addition of silver content on the microstructural variation and antibacterial performance of 2205 duplex stainless steel. The authors evaluate the microstructrures modified by the addition of silver. In addition, the authors show the influence of both microstructures on the antibacterial properties.
In order to improve the manuscript quality, grammatical errors (missing punctuation, incorrect word spacing) should be considered.
According to these comments, my suggestion is to accept the manuscript with minor revisions.
Author Response
19/12/2020
Prof. Dr. Hugo F. Lopez
Editor-in-Chief
Metals
Dear Prof. Dr. Hugo F. Lopez:
We would like to resubmit our revised manuscript, titled " Effect of Ag-rich Nano-Precipitates on the Antibacterial Properties of 2205 Duplex Stainless Steel. " (Manuscript ID: metals-1023440), for consideration for publication in Metals.
The editor and reviewers have recommended major revisions. We have carefully reviewed these suggestions and revised our text accordingly. Please note that those words within manuscript, tables and figures are written with red words for correction. Our responses to each of the reviewers’ comments are detailed below. The details are provided in the manuscript and mark with red for correction.
Comments and Suggestions for Authors:
Reviewer II.
The manuscript entitled “Effect of Ag-rich Nano-Precipitates on Antibacterial Properties of 2205 Duplex Stainless Steel” (metals-1023440)” is an interesting study about the effects of the addition of silver content on the microstructural variation and antibacterial performance of 2205 duplex stainless steel. The authors evaluate the microstructures modified by the addition of silver. In addition, the authors show the influence of both microstructures on the antibacterial properties.
In order to improve the manuscript quality, grammatical errors (missing punctuation, incorrect word spacing) should be considered.
Answer: Thank you for your pointing out. The manuscript has been edited by a professional English editing group to check grammar error. Besides, we have attached the CERTIFICATE OF ENGLISH EDITING.
According to these comments, my suggestion is to accept the manuscript with minor revisions.

Reviewer 3 Report
Before publication in Metals, the authors should take into account some major revision:
[1] The incorporation of Ag nanoparticles has been extensively studied earlier in the community. But the literature survey was not satisfactory. Need to add more to describe the main motivation. [2] Results are not discussed on the basis of previous work reported in the open literature. The lack of discussion makes it difficult to determine the scientific contribution of the present work. [3] The captions to the figures must give full details so that the information can be clearly understood by the reader.
[4] The cross-check report (anti-plagiarism software) found that 31% of the content of the manuscript is taken from other sources. So please reduce this number.
[5] Relevant literature should be cited. The following examples are highly recommended;
[a] Effect of silver on antibacterial properties of stainless steel, Applied Surface Science 256(11):3642–3646.
[b] Antimicrobial Particulate Silver Coatings on Stainless Steel Implants for Fracture Management, materials Science Engineering C. 32 (2012) 1112-1120
[c] Evaluation of Antibacterial Effects of Silver-Coated Stainless Steel Orthodontic Brackets, J Dent (Tehran). 2016 Jan; 13(1): 49–54.\
[d] Recent progress in surface modification of metals coated by plasma electrolytic oxidation: principle, structure, and performance, Progress in Materials Science, 100735
[e] Simultaneous improvement of corrosion resistance and bioactivity of a titanium alloy via wet and dry plasma treatments, Journal of Alloys and Compounds 851, 156840
[f] Effects of concentration of Ag nanoparticles on surface structure and in vitro biological responses of oxide layer on pure titanium via plasma electrolytic oxidation, Applied surface Science .347 (2015) 574-582
Author Response
19/12/2020
Prof. Dr. Hugo F. Lopez
Editor-in-Chief
Metals
Dear Prof. Dr. Hugo F. Lopez:
We would like to resubmit our revised manuscript, titled " Effect of Ag-rich Nano-Precipitates on the Antibacterial Properties of 2205 Duplex Stainless Steel. " (Manuscript ID: metals-1023440), for consideration for publication in Metals.
The editor and reviewers have recommended major revisions. We have carefully reviewed these suggestions and revised our text accordingly. Please note that those words within manuscript, tables and figures are written with red words for correction. Our responses to each of the reviewers’ comments are detailed below. The details are provided in the manuscript and mark with red for correction.
Comments and Suggestions for Authors:
Reviewer III
Before publication in Metals, the authors should take into account some major revision:
[1] The incorporation of Ag nanoparticles has been extensively studied earlier in the community. But the literature survey was not satisfactory. Need to add more to describe the main motivation.
Answer: We appreciate this helpful comment from the reviewer. We have supplemented the description and literature survey in introduction. This information has been supplemented at line 76-91 Page 2 of revised manuscript and marked with red.
[2] Results are not discussed on the basis of previous work reported in the open literature. The lack of discussion makes it difficult to determine the scientific contribution of the present work.
Answer: Thank you for your pointing out. We have supplemented the description which is based on previous work in the discussion. This information has been supplemented at line 226-233 Page 7 of revised manuscript and marked with red.
[3] The captions to the figures must give full details so that the information can be clearly understood by the reader.
Answer: We appreciate this helpful comment from the reviewer. The captions to the figures were revised to make reader understand.
[4] The cross-check report (anti-plagiarism software) found that 31% of the content of the manuscript is taken from other sources. So please reduce this number.
Answer: Thank you for your reminder and suggestion. After revising the manuscript, the anti-plagiarism software (turnitin) showed that 26% of the content of the manuscript is taken from other sources.
[5] Relevant literature should be cited. The following examples are highly recommended;
[a] Effect of silver on antibacterial properties of stainless steel, Applied Surface Science 256(11):3642–3646.
[b] Antimicrobial Particulate Silver Coatings on Stainless Steel Implants for Fracture Management, materials Science Engineering C. 32 (2012) 1112-1120
[c] Evaluation of Antibacterial Effects of Silver-Coated Stainless Steel Orthodontic Brackets, J Dent (Tehran). 2016 Jan; 13(1): 49–54.
[d] Recent progress in surface modification of metals coated by plasma electrolytic oxidation: principle, structure, and performance, Progress in Materials Science, 100735
[e] Simultaneous improvement of corrosion resistance and bioactivity of a titanium alloy via wet and dry plasma treatments, Journal of Alloys and Compounds 851, 156840
[f] Effects of concentration of Ag nanoparticles on surface structure and in vitro biological responses of oxide layer on pure titanium via plasma electrolytic oxidation, Applied surface Science .347 (2015) 574-582
Answer: Thank you for your suggestion. The literature you provided are helpful and meaningful to be cited in this article. This information has been supplemented at Reference 18, 34, 47, 41, 42, 75 Page 12~14 of revised manuscript and marked with red.
Thank you for your consideration. We hope our manuscript is suitable for publication in your journal.
Sincerely,
Professor
Ker-Kong Chen
Department of Dentistry,
Kaohsiung Medical University Hospital,
Kaohsiung 807, Taiwan, ROC.
enamel@kmu.edu.tw

Round 2
Reviewer 3 Report
The authors have addressed all my concerns. It can be accepted now